# Cross-sectional study to assess depression among healthcare workers in Lusaka, Zambia during the COVID-19 pandemic

Sandra Simbeza,[1] Jacob Mutale,[1] Musunge Mulabe,[1] Lazarus Jere,[1] Chama Bukankala,[1] Kombatende Sikombe [ORCID],[2] Izukanji Sikazwe,[1] Carolyn Bolton-Moore,[1,3] Aaloke Mody [ORCID],[4] Elvin H Geng,[4] Anjali Sharma,[1] Laura K Beres,[5] Jake M Pry [ORCID] [2,6]

LKB and JMP contributed equally.

For numbered affiliations see end of article.

**Correspondence to**
Dr Jake M Pry;
jmpry@ucdavis.edu

## ABSTRACT

**Objectives** We sought to assess depression among healthcare workers (HCWs) in the context of COVID-19 in Lusaka Province, Zambia.

**Design** This cross-sectional study is nested within a larger study, the Person-Centred Public Health for HIV Treatment in Zambia (PCPH), a cluster-randomised trial to assess HIV care and outcomes.

**Setting** The research was conducted in 24 government-run health facilities from 11 August to 15 October 2020 during the first wave of the COVID-19 pandemic in Lusaka, Zambia.

**Participants** We used convenience sampling to recruit HCW participants who were previously enrolled in the PCPH study, had more than 6 months' experience working at the facility and were voluntarily willing to participate.

**Primary outcome measures** We implemented the well-validated 9-question Patient Health Questionnaire (PHQ-9) to assess HCW depression. We used mixed-effects, adjusted Poisson regression to estimate the marginal probability of HCWs experiencing depression that may warrant intervention (PHQ-9 score ≥5) by healthcare facility.

**Results** We collected PHQ-9 survey responses from 713 professional and lay HCWs. Overall, 334 (46.8%, 95% CI 43.1%, 50.6%) HCWs recorded a PHQ-9 score ≥5, indicating the need for further assessment and potential intervention for depression. We identified significant heterogeneity across facilities and observed a greater proportion of HCWs with symptoms of depression in facilities providing COVID-19 testing and treatment services.

**Conclusions** Depression may be a concern for a large proportion of HCWs in Zambia. Further work to understand the magnitude and aetiologies of depression among HCWs in the public sector is needed to design effective prevention and treatment interventions to meet the needs for mental health support and to minimise poor health outcomes.

## STRENGTHS AND LIMITATIONS OF THIS STUDY

⇒ These data represent important insights regarding the state of mental wellness among a large sample of healthcare workers (HCW) during a public health crisis in a resource-limited setting, where mental wellness, such as depression, is often not measured nor documented.

⇒ We include a large number (n=24) of facilities to appreciate the heterogeneity in symptoms of depression among mental health in the sampled facilities in Lusaka Province of Zambia.

⇒ An important limitation is lack of data on demographics of HCW respondents due to concern about stigma around mental wellness and potential to identify individuals given inclusion of clinics with small staff.

⇒ Without pre-COVID-19 estimates of depression among HCWs in Lusaka, we are not able to show the association between mild-severe depression with the pandemic; however, we show that during the pandemic depression was high and attention to this population is justified to ensure a healthy HCW workforce.

⇒ We employed convenience sampling in selecting participants which may limit the representativeness of study results to the wider population of HCWs.

## INTRODUCTION

COVID-19 has caused a substantial global health hardship and the position of healthcare workers (HCWs) on the frontlines of the public health response places them at great risk for negative effects on health and well-being.[1] In addition to excess occupational hazard of contracting COVID-19, their experience as caregivers increases their risk of developing mental health disorders such as anxiety, depression, trauma, insomnia and stress, which may lead to poor physical and psychological well-being.[2] A healthy workforce is critical to effectively managing and mitigating COVID-19, as well as providing continuity of high-quality care for other chronic and acute health conditions.[3 4] Even

**Table 1** Participant characteristics

| Factor | Level | n (%) |
|---|---|---|
| N | | 713 |
| Facility | 1 | 30 (4.2) |
| | 2 | 30 (4.2) |
| | 3 | 30 (4.2) |
| | 4 | 30 (4.2) |
| | 5 | 30 (4.2) |
| | 6 | 30 (4.2) |
| | 7 | 30 (4.2) |
| | 8 | 30 (4.2) |
| | 9 | 30 (4.2) |
| | 10 | 30 (4.2) |
| | 11 | 30 (4.2) |
| | 12 | 27 (3.8) |
| | 13 | 27 (3.8) |
| | 14 | 30 (4.2) |
| | 15 | 30 (4.2) |
| | 16 | 30 (4.2) |
| | 17 | 30 (4.2) |
| | 18 | 30 (4.2) |
| | 19 | 30 (4.2) |
| | 20 | 30 (4.2) |
| | 21 | 30 (4.2) |
| | 22 | 30 (4.2) |
| | 23 | 29 (4.1) |
| | 24 | 30 (4.2) |
| Clinic population (category) | Small | 147 (20.6) |
| | Medium | 269 (37.7) |
| | Large | 297 (41.7) |
| Month of survey | August 2020 | 205 (28.8) |
| | September 2020 | 496 (69.6) |
| | October 2020 | 12 (1.7) |

prior to COVID-19, countries in sub-Saharan Africa faced limited medical infrastructures, supplies, and an overburdened workforce, which challenged HCW well-being and the provision of quality healthcare.[5 6] These challenges, along with pandemic-sensitive barriers such as limited access to personal protective equipment (PPE), further exacerbated HCW stress and vulnerability. While HCWs in resource-limited settings have demonstrated resilience, poor mental health is likely to compromise their ability to make decisions, as well as impact patient interactions.[7] Mental health services, like other health resources, are limited, with few trained mental health providers.[8] Understanding the mental health and well-being among HCWs can catalyse interventions to provide treatment and improve the healthcare facility environment for the HCWs and patients.[8]

Several studies have been conducted to assess the mental wellness impact of the pandemic among HCWs; however, the majority of these studies focus on the continent of Asia and very little data are available for HCWs in Africa.[9–11] Limited data from Kenya and Ethiopia provide evidence that the prevalence of mental disorders such as depression, insomnia and stress was higher among those HCWs caring for patients with COVID-19, or in areas of higher infection prevalence compared with those working with non-COVID-19 patients or less.[2 12–16] The most prevalent reported mental health conditions among HCWs are depression, insomnia and anxiety.[17] Characterisation of the state of mental wellness during the COVID-19 pandemic in sub-Saharan African countries among HCWs, specifically in Zambia, remains incomplete.[12 18]

As part of a larger patient-centred care study, we collected facility-level measures to understand the context of care at 24 study sites. As a part of establishing care context we assessed mental health, specifically depression, using the 9-question Patient Health Questionnaire (PHQ-9) in the context of COVID-19 among HCWs in Lusaka Province, Zambia from 11 August 2020 to 15 October 2020.[19]

## METHODS

### Setting

This cross-sectional study of HCW depression is nested within a larger study, the Person-Centred Public Health for HIV Treatment in Zambia (PCPH), a cluster-randomised trial to assess HIV care and outcomes running from August 2018 to November 2021 across 24 government-funded health facilities in Lusaka Province (Pan African Clinical Trials Registry number: PACTR202101847907585). All PHQ-9 responses were collected between 11 August 2020 and 15 October 2020 among participating facilities that offered HIV care services and varied in size and location. Facilities were assigned codes to protect the identity of the HCW participants.

### Study population

We developed a cohort of HCWs who primarily provide HIV care at one of the 24 health facilities in the PCPH study. This was done by compiling all contact details from the HCWs who had, at the time of the introduction of the depression study, participated in the HCW survey component of the original PCPH study. The PCPH sample comprised both professional HCWs, including nurses, pharmacists, midwives, medical doctors and radiographers, and lay HCWs including treatment supporters and general workers. To augment participation in the PHQ-9 data collection, we expanded the PHQ-9 sample to include any HCW willing to participate and had ≥6 months' working experience at the study facility. We informed the facility in charge of the PHQ-9 study opportunity who communicated it to staff members.

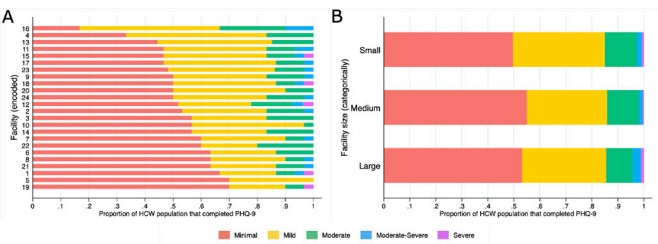

**Figure 1** (A) Stacked bar chart of proportion population by 9-question Patient Health Questionnaire (PHQ-9) score category by healthcare facility. (B) Stacked bar chart of proportion population by PHQ-9 score by facility population size category. HCW, healthcare worker.

## Measurements

Trained study research assistants visited healthcare facilities and discussed the study opportunity in person with available staff members. Those who expressed interest were screened for eligibility, offered the opportunity to verbally consent and participate immediately in the PHQ-9 study. As a part of the PCPH adaptation to the COVID-19 pandemic, we implemented the 9-question Patient Health Questionnaire (PHQ-9) to screen for presence and severity of depression. The PHQ-9 has been previously used in Zambia to screen for likelihood of depression,[19] and has been validated in Tanzania and South Africa which are similar settings as Zambia.[19 20] The PHQ-9 instrument was translated from English into Nyanja and Bemba, the most commonly spoken Zambian languages in Lusaka Province, where the survey was conducted.[21] Depending on the participant's preference, the standard PHQ-9 survey was self-administered on paper or surveyor administered by trained study research assistants in the participant's preferred language. Potentially identifying information such as age, sex and HCW cadre was not collected from respondents to protect privacy. Facility populations were categorised as small (<40 000 clients/year), medium (40 000–99 999 visits/year) and large (≥100 000 visits/year) as recorded in 2019.

**Table 2** Proportion of the analysis population by PHQ-9 score category with 95% CIs (N=713)

| Variable | n | Proportion (%) | 95% CI |
|---|---|---|---|
| PHQ-9 score ≥5 | 334 | 46.8 | 43.1, 50.6 |
| Mild | 231 | 32.4 | 29.0, 36.0 |
| Moderate | 81 | 11.4 | 9.1, 13.9 |
| Moderate-severe | 17 | 2.4 | 1.4, 3.8 |
| Severe | 5 | 0.7 | 0.2, 1.6 |

CI, confidence interval; PHQ-9, 9-question Patient Health Questionnaire.

## Analysis

We followed the standard 27-point scoring system for the PHQ-9 to identify participants with scores consistent with minimal (0–4), mild (5–9), moderate (10–14), moderately severe (15–19) and severe depression (20-27).[19–26] Our primary outcome was mild depression or greater (PHQ-9 score ≥5) as this level of depression warrants additional clinical follow-up.[19] We developed frequency tables and used a bar graph to illustrate the distribution of PHQ-9 scores by healthcare facility. We developed scatter plots of adjusted marginal probability with 95% CIs to illustrate the probability that an HCW will have a PHQ-9 score ≥5 by healthcare facility. We used mixed-effects Poisson regression to estimate the prevalence ratios for those with mild depression allowing random effects at the facility level and measured fixed effects for month of survey and clinic size category. Facilities were categorised by client population estimates as: small (<40 000 clients), medium (40 000–100 000 clients) and large (>100 000 clients).

## Patient and public involvement

The parent study focused primarily on improving the patient experience at routine clinic visits through HCW training, mentorship and audit and feedback. The research approach and content is based on participatory research with HCWs, human-centred design workshops and stakeholder collaborations.[22–28] This research guided us to include measures of HCW satisfaction and well-being. In addition to an HCW experience measure throughout PCPH, the advent of COVID-19 led us to include an assessment of HCW depression. Patients were not directly involved in the design nor the recruitment of this substudy assessing the depression levels of HCWs. The findings of this study as well as the parent study will be shared with the Zambian Ministry of Health as well as the study facilities.

## RESULTS

A total of 713 HCWs from 24 facilities across Lusaka and Chongwe districts in the Lusaka Province were included in the analysis data set (table 1). The majority (69.6%) of the PHQ-9 survey responses were collected in September 2020 (table 1). The largest proportion of the responses was collected at facilities serving a large client population (41.7%), followed closely by facilities serving a medium-sized client population (37.7%) (table 1). Responses for PHQ-9 questions were largely complete with <1% of responses missing.

Of the 713 responses, 231 (32.4%, 95% CI 29.0%–36.0%) reported mild depression (PHQ-9 score 5–9), and 81 (11.4%, 95% CI 9.1%–13.9%) reported moderate depression (PHQ-9 score 10–14). A total of 81 (11.4%) respondents had a PHQ-9 score corresponding to moderate depression across all but one facility, 17 (2.4%, 95% CI 1.4%–3.8%) respondents had PHQ-9 scores consistent with moderately severe depression across four facilities and 5 (0.7%, 95% CI 0.2%–1.6%) HCWs had

**Table 3** Adjusted Poisson regression results (N=713)

| Covariate | Level | aPR | 95% CI | P value |
|---|---|---|---|---|
| Clinic population (category) | Small | 1.12 | 0.95, 1.33 | 0.170 |
| | Medium | 1.00 (ref) | Ref | Ref |
| | Large | 1.04 | 0.81, 1.33 | 0.763 |
| Month of survey | August | 1.00 (ref) | Ref | Ref |
| | September | 1.14 | 0.94, 1.38 | 0.193 |
| | October | 0.98 | 0.54, 1.79 | 0.945 |

Adjusted for survey week and facility.
aPR, adjusted prevalence ratio; CI, confidence interval.

PHQ-9 scores consistent with severe depression across five different facilities (figure 1, online supplemental figure S1, table 2).

Though we observed facility-level mental health heterogeneity in the proportion of minimal and mild depression scores across clinics, the proportion of scores consistent with moderate depression remains relatively stable across facilities (figure 1).

Mixed-effects adjusted Poisson regression model did not reveal clinic size or month of survey as a significant predictor of PHQ-9 score ≥5 (table 3).

We illustrate significant heterogeneity in the marginal probability of experiencing greater than minimal depression across facilities (figure 2). Notably, the highest marginal probability of HCWs with PHQ-9 score ≥5 was observed in a facility serving as a COVID-19 treatment centre.

## DISCUSSION

We found that a large proportion of the HCW population had a PHQ-9 score ≥5, indicating a need for follow-up to assess and improve mental well-being, 46.8% (95% CI 43.1%, 50.6%). Variation in depression outcomes ranged

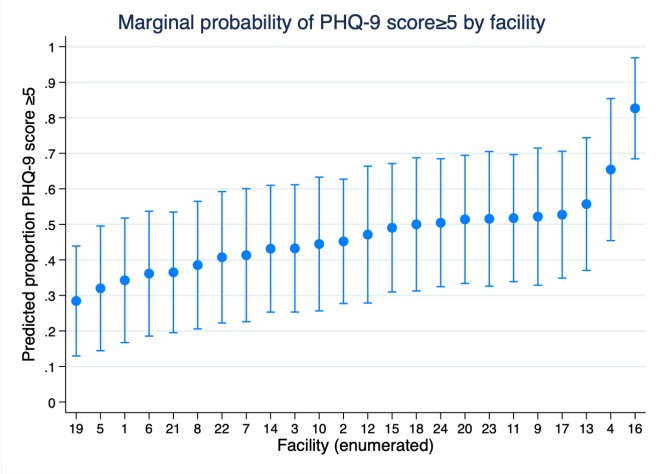

**Figure 2** Marginal probability of healthcare worker with greater than minimal depression (score ≥5) per 9-question Patient Health Questionnaire (PHQ-9) score by clinic.

from 82.7% (95% CI 68.5%, 96.9%) at one of the two large COVID-19 treatment facilities to 28.4% (95% CI 13.0%, 43.9%) at a medium-sized healthcare facility.

Our study shows that HCWs working at the COVID-19 treatment facility had a higher marginal probability of experiencing mild to moderately severe depression. This is consistent with past studies showing that frontline HCWs working in clinics managing diseases considered to be highly infectious, such as COVID-19 treatment centres, were more prone to developing depression and other mental disorders compared to their counterparts in other departments.[12] In addition, results of our study demonstrate that HCWs have experienced symptoms of depression during the onset of the COVID-19 pandemic in Zambia, which is aligned with the findings of similar studies using the PHQ-9 in other parts of the world where a pooled prevalence of mild depression was found to be 36.1% (95% CI 31.3%–41.0%).[29] Studies conducted in Ethiopia among different cadres of HCWs show that mental disorders which include depression are prevalent among HCWs during the COVID-19 pandemic, with one study from Ethiopia identifying approximately 48% prevalence of greater than minimal depression, consistent with our Zambian estimates.[30 31] Similarly, global studies indicate that there is a risk of HCWs experiencing mental health disorders during the pandemic.[4 29 30 32] The difficult conditions to which HCWs may be exposed including extended working hours, risk of exposure to the disease, increased workload, concerns about transmitting the infection to their family members, reduced social connectedness and limited resources to care for their patients may amplify poor mental well-being.[1 33–35]

Further research is needed to understand the heterogeneity in proportions of HCWs with depression. It may be associated variation in HCW access to resources, such as PPE, hand hygiene station/station supplies at clinic entrance and knowledge about the COVID-19 response in Zambia. While the response to the COVID-19 pandemic was standardised, to a certain extent, by guidance from the provincial and zonal levels, the culture and leadership unique to each facility might have played a key role in the HCW experience, contributing to PHQ-9 score distribution differences across facilities.

As we continue to recognise the mental health services gap across many populations in resource-limited settings we give evidence here to support prioritisation of HCWs, especially during public health shocks/emergencies like that presented by COVID-19. Presence of depression among HCWs could lead to poor health outcomes for the HCW workforce and have a sort of 'knock-on' effect negatively impacting the quality of care provided to patients.[36 37] Interventions like Friendship Bench piloted in neighbouring Zimbabwe designed to encourage positive coping mechanism among HCWs and build a working environment that provides empathy and compassion towards staff may be an efficient option to provide mental wellness support.[38 39] Furthermore, system-based interventions should also be encouraged such as change

in working culture and reduction in possible system contributors to HCW stress that could lead to depression. Increasing mobile technology availability may further allow for the use of mobile health-based mental wellness services leveraging the framework presented by Osei and Mashamba-Thompson for low and middle-income countries.[40 41] Low-cost intervention packages used for patients can be adapted for HCWs and integrated into system-based interventions. They include routine screening for early detection, mental wellness education, problem solving, use of antidepressants, and cognitive–behavioural therapy which can be delivered successfully by trained lay HCWs.[42 43]

## Limitations

This study had some potential limitations. First, we did not collect participant demographics such as age, sex and marital status to protect participant privacy, especially at small clinics, where staff are few. Data for these potential confounders might help to better understand associations with depression. Second, we employed convenience sampling in selecting participants. This sampling approach may limit the representativeness of the study results to the wider population of HCWs. Finally, without pre-COVID-19 estimates of depression among HCWs in Lusaka, we are not able to show the association between mild-severe depression with the pandemic; however, we show that during the pandemic the depression was high and attention to this population is justified to ensure a healthy HCW workforce. Additionally, though potentially higher than a non-pandemic baseline, these results may be useful as subsequent measures of depression and mental wellness are collected among HCWs.

## CONCLUSIONS

Depression is a common public health problem; our study demonstrates that HCWs in Zambia may suffer from a high prevalence of depressive symptoms that will require additional clinical follow-up. Routine mental health wellness is important to better understand the role that the COVID-19 pandemic may have had on depression among HCWs in Zambia. Furthermore, support for HCW mental wellness can serve to accelerate destigmatising mental health issues and improve the quality of care provided across healthcare centres in Zambia.

**Author affiliations**
[1]Research Department, Center for Infectious Disease Research in Zambia, Lusaka, Zambia
[2]Implementation Science Unit, Research Department, Center for Infectious Disease Research in Zambia, Lusaka, Zambia
[3]Division of Infectious Diseases, The University of Alabama at Birmingham School of Medicine, Birmingham, Alabama, USA
[4]Division of Infectious Diseases, Washington University in St Louis, St Louis, Missouri, USA
[5]Department of International Health, Johns Hopkins Bloomberg School of Public Health, Baltimore, Maryland, USA
[6]Public Health Sciences, University of California Davis School of Medicine, Sacramento, California, USA

**Acknowledgements** We thank the Zambian Ministry of Health for their leadership in ensuring that those in care continue to receive life-saving HIV treatment. We also thank the healthcare workers for their participation in the PHQ-9 assessment as well as their faithful delivery of care to patients in the face of the many challenges presented by the COVID-19 pandemic. To the clients seen at our study facilities, we extend our sincere thanks for your cooperation and patience.

**Contributors** SS: data collection, data framing and manuscript writing. JM, KS and MM: data collection and data curation. LJ and CB: data framing. ITS, CB-M and EG: conceptualisation. AM: manuscript review/editing. AS: conceptualisation and manuscript review/editing. LKB: conceptualisation, data analysis, data framing and manuscript writing. JMP: guarantor, data analysis lead, data framing and manuscript writing.

**Funding** Research reported in this publication was supported by the Bill & Melinda Gates Foundation (grant number: OPP1166485), the National Institute of Mental Health of the National Institutes of Health (award number: F31MH109378 (LKB)) and the National Institute of Allergy and Infectious Diseases of the National Institutes of Health (award number: K24 AI134413 (EG)).

**Competing interests** None declared.

**Patient and public involvement** Patients and/or the public were involved in the design, or conduct, or reporting, or dissemination plans of this research. Refer to the Methods section for further details.

**Patient consent for publication** Not applicable.

**Ethics approval** This study was approved as part of the larger PCPH study by the University of Zambia Biomedical Ethics Committee and the University of Alabama at Birmingham Institutional Review Board, and the National Health Research Authority in Zambia. We obtained waiver of written informed consent and obtained verbal consent from participants prior to administering the survey.

**Provenance and peer review** Not commissioned; externally peer reviewed.

**Data availability statement** Data are available upon reasonable request. De-identified data used for this analysis can be accessed via the Dryad data repository at http://datadryad.org/ with doi https://doi.org/10.25338/B8S646.

**ORCID iDs**
Kombatende Sikombe http://orcid.org/0000-0002-8187-8661
Aaloke Mody http://orcid.org/0000-0003-3787-365X
Jake M Pry http://orcid.org/0000-0001-6312-4420

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
