## [Reviewer comments · BMJ Open]

ARTICLE DETAILS

TITLE (PROVISIONAL)	A cross-sectional study to assess depression among health care workers in Lusaka, Zambia during the COVID-19 pandemic
AUTHORS	Simbeza, Sandra; Mutale, Jacob; Mulabe, Musunge; Jere, Lazarus; Bukankala, Chama; Sikombe, Kombatende; Sikazwe, Izukanji; Bolton-Moore, Carolyn; Mody, Aaloke; Geng, Elvin; Sharma, Anjali; Beres, Laura K; Pry, Jake

VERSION 1 – REVIEW

REVIEWER	Cicek, Ilhan Batman Üniversitesi Merkez Kampüsü
REVIEW RETURNED	08-Nov-2022

GENERAL COMMENTS	Thank you for considering me to review the paper entitled " Mental Wellbeing of Health Care Workers in Lusaka, Zambia During the COVID-19 Pandemic". Although I consider the paper has some potential implications for research and practice, I want to address several concerns. Abstract: Well written. Introduction: Well written. Please update the literature with recants papers such as; 1. Ceri, V., & Cicek, I. (2021). Psychological well-being, depression and stress during COVID-19 pandemic in Turkey: A comparative study of healthcare professionals and non-healthcare professionals. Psychology, Health & Medicine, 26(1), 85-97.2. Yıldırım, M., Çiçek, İ., & Şanlı, M. E. (2021). Coronavirus stress and COVID-19 burnout among healthcare staffs: The mediating role of optimism and social connectedness. Current Psychology, 40(11), 5763-5771.3. Yıldırım, M., & Çiçek, İ. (2022) Optimism and pessimism mediate the association between parental coronavirus anxiety and depression among healthcare professionals in the era of COVID-19, Psychology, Health & Medicine, 27:9, 1898-1906, DOI: 10.1080/13548506.2021.1966702 Method Well-written Results: Good designed Discussion: Well written. Please update the literature. Conclusion: Good.
---

	Limitations: Good References: References must be checked thoroughly.
--	---

REVIEWER	Tariku, Mandaras Haramaya University, psychiatry
REVIEW RETURNED	17-Nov-2022

GENERAL COMMENTS	First of all I thank you for providing a chance to reviewing this paper. Abstract Title: it is not clear, mental wellbeing or depression or common mental disorders? Because you used PHQ-9, a tool for depression. Your title need improvements. "Depression among....." Objective: need consistency with your tittle. Mental wellbeing and mental health problems or common mental disorder are quite different. Read the literatures. Methods: what kind of analysis method did you used? Incorporate in the methods. Results: "as well as preventative 16 interventions, to minimize the possibility of poor health outcomes" this phrase indicates the recommendations. Therefore, after conclude your results insert this phrase. Remove from the results. In addition, if you have any variable which are significant predictor for depression you can narrate it in the results. Conclusion: conclude your finding based on your results. Some words also need consistent" wellbeing or depression or mental health disorder or mental health problems" in all over your document. Introduction Citation line 41. Methods: why poisson regression? Did you assessed predictors? How did you see the profession of HCW? Frontline or not? Unit of HCW (ICU, OPD, ward)? What does mean lay HCW?
---

VERSION 1 – AUTHOR RESPONSE

Reviewer: 1

Comments to the Author:

Nice work. I congratulate you. However, especially the introduction and discussion sections need to be updated and revised.

Thank you for your positive assessment of the work. We engaged additional literature in the introduction and discussion, including three papers from Ceri et al, Yildirim et al., and Yildirim et al.

Reviewer: 2

Comments to the Author:

First, I thank you for providing a chance to reviewing this paper.

Response: Thank you for your time and expertise in reviewing our paper.

Abstract

Title: it is not clear, mental wellbeing or depression or common mental disorders? Because you used PHQ-9, a tool for depression. Your title need improvements. "Depression among....."

Response: We have revised the title according to the feedback provided: The title has been revised to "A cross sectional survey to assess depression among Health Care Workers in Lusaka, Zambia

during the COVID 19 pandemic.”

Objective: need consistency with your title. Mental wellbeing and mental health problems or common mental disorder are quite different. Read the literatures.

Response: We appreciate the feedback, we have revised the title as indicated above to “A cross-sectional survey to assess depression among Health Care Workers in Lusaka, Zambia during the COVID 19 pandemic.”

Methods: what kind of analysis method did you use? Incorporate in the methods.

Response: Thank you. We used descriptive analyses (e.g., frequency tables and bar charts), scatter plots of adjusted marginal probability, and mixed effects Poisson regression to estimate prevalence ratios for those with mild depression allowing random effects at the facility level and measured fixed effects for month of survey and clinic size category. These are included in the abstract and methods section.

Results: “as well as preventative 16 interventions, to minimize the possibility of poor health outcomes” this phrase indicates the recommendations. Therefore, after conclude your results insert this phrase. Remove from the results. In addition, if you have any variable which are significant predictor for depression you can narrate it in the results.

Response: We have made the revision according to the feedback provided. We do not however, have any additional variables which are significant predictor of depression such as sex, age, and type of cadre for Health Care Workers. These were not collected to protect the privacy of our study participants. We acknowledge this as a limitation to the study and it has been highlighted in the limitations sections.

Conclusion: conclude your finding based on your results. Some words also need consistent” wellbeing or depression or mental health disorder or mental health problems” in all over your document.

Response: We have revised the conclusion based on the feedback. We have also revised the text in the document and maintained the word depression for consistency and to be reflective of the tool (Patient Health Questionnaire -PHQ-9) that we used assess depression among Health Care Workers, in Lusaka, Zambia

Introduction Citation line 41.

Response: We have included the citation to the sentence in the introduction.

Methods: why poisson regression? Did you assess predictors? How did you see the profession of HCW? Frontline or not? Unit of HCW (ICU, OPD, ward)? What does mean lay HCW?

Response:

- We used Poisson regression because our data is count data and this model would be ideal for predicting the frequency of occurrence of event – in this case depression.
- Assessing predictors: We looked at associations between facility and survey date. We did not collect individual-level demographics, including specific department or cadre (e.g., frontline/OPD) or link lay versus professional status to individual responses to protect privacy in this sensitive research topic. We note this as a limitation in our manuscript. We
- We revised our study population section to better specify professional versus lay HCW roles: “The PCPH sample was comprised of both professional HCWs including nurses, pharmacists, midwives, medical doctors, radiographers, and lay HCWS including treatment supporters and general workers.”

Editor(s) Comments to Author (if any):

-Please revise the title of your manuscript to include the research question, study design and setting. This is the preferred format of the journal.

Response: We have revised the title of the manuscript according to feedback provided. The revised title is “A cross sectional survey to assess depression among Health Care Workers in Lusaka, Zambia during the COVID 19 pandemic.”

-Please ensure that your abstract is formatted according to our Instructions for Authors:

https://nam12.safelinks.protection.outlook.com/?url=http%3A%2F%2Fbmjopen.bmj.com%2Fpages%2Fauthors%2F%23research&data=05%7C01%7Cjmpy%40ucdavis.edu%7C435839f7c1064d389b4208dad16b7d4c%7Ca8046f6466c04f009046c8daf92ff62b%7C0%7C0%7C638052556837102378%7CUnknown%7CTWFpbGZsb3d8eyJWIjoiMC4wLjAwMDAiLCJQIjoiV2luMzliLCJBTiI6Ikl1haWwiLCJXVCI6Mn0%3D%7C3000%7C%7C&sdata=v6pHxNAefXDjKv9NXGff1GtWLeFVANGg9kAY%2BVqMZvw%3D&reserved=0

Response: We have revised the outline of the abstract according to the guidelines provided.

-Please revise the 'Strengths and limitations of this study' section of your manuscript (after the abstract). This section should contain up to five short bullet points, no longer than one sentence each, that relate specifically to the methods. The novelty, aims, results or expected impact of the study should not be summarized here.

Response: We have included the strengths and limitations of our manuscript according to the feedback.

End itemized review response.

VERSION 2 – REVIEW

REVIEWER	Cicek, Ilhan Batman Üniversitesi Merkez Kampüsü
REVIEW RETURNED	03-Jan-2023

GENERAL COMMENTS	Congratulations for this valuable and impressive work.
--

REVIEWER	Tariku, Mandaras Haramaya University, psychiatry
REVIEW RETURNED	01-Feb-2023

GENERAL COMMENTS	I thank you for giving the chance to reviewing this paper." Dear authors of this manuscript, I have read your paper in detailed. I have observed several studies conducted in this area. Your paper missed the predictors that affect the depression in this population. Your references are also not updated. In addition, I have missed the new variables or concepts that your study added for the scientific worlds.
--

VERSION 2 – AUTHOR RESPONSE

Reviewer: 1

Dr. Ilhan Cicek, Batman Üniversitesi Merkez Kampüsü

Comments to the Author:

Congratulations for this valuable and impressive work.

Many thanks, again, for your positive assessment of the work.

Reviewer: 2

Mr. Mandaras Tariku, Haramaya University

Comments to the Author:

I thank you for giving the chance to reviewing this paper.” Dear authors of this manuscript, I have read your paper in detailed. I have observed several studies conducted in this area. Your paper missed the predictors that affect the depression in this population. Your references are also not updated. In addition, I have missed the new variables or concepts that your study added for the scientific worlds.

Thank you for your time and feedback. Predictors considered include all available data. We have updated citations to include Yadeta et al. and Fond et al. We selected these studies as they focus on the mental wellbeing of healthcare workers, our study population. Yadeta et al. found the prevalence of greater than minimal depression to be approximately 48% in a cross-section survey of Ethiopian healthcare workers. This is consistent with our Zambian estimate of 46.8%. The purpose of this paper is to describe the prevalence of depression among HIV healthcare workers in Zambia during the COVID-19 pandemic. To our knowledge, this is the only study assessing depression amongst healthcare workers in Zambia during the COVID-19 pandemic. This adds a critical perspective from Zambia and, by extension, the Southern African region, to the mental health literature and the COVID-19 literature.

End itemized review response.